# Activation of Alternative Bilirubin Clearance Pathways Partially Reduces Hyperbilirubinemia in a Mouse Model Lacking Functional Ugt1a1 Activity

**DOI:** 10.3390/ijms231810703

**Published:** 2022-09-14

**Authors:** Bhaswati Banerjee, Olayemi Joseph Olajide, Giulia Bortolussi, Andrés F. Muro

**Affiliations:** Mouse Molecular Genetics Group, ICGEB, Padriciano, 99, 34149 Trieste, Italy

**Keywords:** liver, constitutive androstane receptor (CAR), bilirubin-degradation pathways, TCPOBOP

## Abstract

Bilirubin is a heme catabolite and Ugt1a1 is the only enzyme involved in the biological elimination of bilirubin. Partially functional or non-functional Ugt1a1 may result in neuronal damage and death due to the accumulation of unconjugated bilirubin in the brain. The understanding of the role of alternative bilirubin detoxification mechanisms that can reduce bilirubin toxicity risk is crucial for developing novel therapeutic strategies. To provide a proof-of-principle showing whether activation of alternative detoxification pathways could lead to life-compatible bilirubin levels in the absence of Ugt1a1 activity, we used Ugt1^−/−^ hyperbilirubinemic mice devoid of bilirubin glucuronidation activity. We treated adult Ugt1^−/−^ mice with TCPOBOP, a strong agonist of the constitutive androstane receptor (CAR). TCPOBOP treatment decreased plasma and liver tissue bilirubin levels by about 38%, and resulted in the transcriptional activation of a vast array of genes involved in bilirubin transport and metabolism. However, brain bilirubin level was unaltered. We observed ~40% degradation of bilirubin in the liver microsomes from TCPOBOP treated Ugt1^−/−^ mice. Our findings suggest that, in the absence of Ugt1a1, the activation of alternative bilirubin clearance pathways can partially improve hyperbilirubinemic conditions. This therapeutic approach may only be considered in a combinatorial manner along with other treatments.

## 1. Introduction

Bilirubin is the end product of heme catabolism and originates primarily from haemoglobin of red blood cells and from various heme- containing proteins. Bilirubin is generated by two-step enzymatic reactions that take place in the cells of the reticuloendothelial system, notably the spleen. Other cells include phagocytes and the Kupffer cells of the liver [1,2]. Firstly, the heme group is cleaved by the microsomal heme-oxigenase 1 (HO-1) to generate biliverdin, which is further reduced to the orange-yellow pigment known as bilirubin or unconjugated bilirubin (UCB) by the enzyme biliverdin reductase (BVR) [3].

UCB is a highly hydrophobic molecule with high affinity for lipid-rich tissues and is transported in the bloodstream bound to albumin. UCB is taken up into the hepatocytes from the liver sinusoids either by passive diffusion or by receptor-mediated endocytosis [4]. UCB undergoes glucuronic acid conjugation in the endoplasmic reticulum of the hepatocytes by the enzyme uridine diphospho-glucuronosyltransferase 1a1 (UGT1a1) [5]. This process of conjugation is mandatory to increase its aqueous solubility, allowing its proper excretion into the bile fluid. Upon glucuronidation, conjugated bilirubin (CB) can pass into the bile majorly via the active transporter, multidrug resistance protein Mrp2 (ABCC2) [6]. A portion of conjugated bilirubin is transported into the sinusoids and portal circulation by MRP3 (ABCC3) [6]. Reports suggest that the uptake of conjugated and unconjugated bilirubin could be also mediated by the members of the organic anion transporting polypeptide (OATP) family, particularly OATPB1B1 and OATPB1B3 [7]. The bile is further released into the intestine and CB is converted to the colorless pigments, urobilinogens. These pigments are further converted to urobilins and stercobilins and are eliminated through urine and feces, respectively [4]. 

Defect or alteration in the bilirubin metabolism pathway results in conjugated and/or unconjugated hyperbilirubinemia. Unconjugated hyperbilirubinemia can result from genetic or non-genetic causes. Genetic defects in the UGT1a1 gene result in Crigler-Najjar syndrome types I and II (CNSI and CNSII), which are associated with complete or partial absence of bilirubin glucuronidation activity, respectively [8]. Neonatal hyperbilirubinemia results from the delayed induction of the glucuronidation system after birth, and moderate UCB levels are considered beneficial due to the antioxidant properties of bilirubin [9]. It is normally self-resolved a few days after birth and it is commonly treated with phototherapy (PT) [10]. However, bilirubin encephalopathy may be seen in severe cases of uncontrolled neonatal jaundice and in CNS patients with toxic spikes of unconjugated hyperbilirubinaemia, which may result in permanent brain damage and death by kernicterus [11]. Uncontrolled cases of neonatal hyperbilirubinemia are treated with intensive phototherapy and eventually exchange transfusion. Patients with CNSI receive up to 12–14 h/day of life long PT, but it does not completely prevent the risk of potential permanent brain damage, leaving liver transplantation as the only therapeutic option [3,12,13]. Thus, the investigation of alternate mechanisms of bilirubin detoxification could aid the development of novel therapeutic strategies for the prevention of hyperbilirubinemia.

The principal pathway for bilirubin clearance mediated by Ugt1a1 is well defined, but other mechanisms of bilirubin degradation have also been reported as suggesting that bilirubin could be partially converted to oxidised metabolites by cytochrome P450 (Cyp) 1 and 2 enzymes [14,15,16,17,18]. These enzymes are reported to play a role in the compensatory pathway of bilirubin metabolism in the complete absence of UGT1a1 activity [16,17,19].

Different approaches have been studied to find out efficient pharmacological options for bilirubin clearance, including the manipulation of metabolic pathways, and key transcription factors such as the nuclear xenobiotic receptor constitutive androstane receptor (NR1I3, CAR) and other transcription factors [20,21,22,23,24,25]. The role of CAR, a nuclear hormone receptor, in the detoxification of bilirubin has been widely studied in in vivo and in vitro experiments [21,23,26]. CAR stimulation results in a coordinated action of bilirubin metabolizing and detoxifying enzymes [27], and in vivo studies have shown that the activation of the bilirubin clearance pathway components is absent in CAR-null mice [26]. Available reports indicate that CAR seems to play the most important role in the regulation of bilirubin metabolism [23,26].

However, the studies related to CAR modulation have been performed in animals having wild type or reduced Ugt1a1 activity [23,27,28], while the real efficacy of these mechanisms in the complete absence of UGT1a1 activity has not yet been determined. Thus, we tested the efficacy of pharmacologic activation of CAR, the major target for the bilirubin clearance pathway, in reducing hyperbilirubinemic conditions in a clinically relevant mouse model of genetically induced hyperbilirubinemia generated in our lab [29,30]. These mice lack functional Ugt1a1 protein and enzyme activity due to a targeted null mutation in Ugt1a exon 4, developing severe hyperbilirubinemia soon after birth. Untreated mutant mice die within the first 2 weeks of life. Treatment with phototherapy for 15 days fully rescues mortality and they reach adulthood despite relatively high blood bilirubin level throughout their lifespan [29,30]. Since these animals do not express Ugt1a1, they fit as the best model to study the alternative pathways for bilirubin clearance. 

In the present study, we administered mice with the CAR ligand 1,4-bis-[2-(3,5-dichlorpyridyloxy)]benzene (TCPOBOP) [31] in the absence of Ugt1a1 activity. We hypothesized that this treatment would strongly activate the otherwise mild downstream molecular pathways involved in bilirubin detoxification and exportation pathways, and shed light on their real role and effects in bilirubin clearance in vivo.

## 2. Results

### 2.1. Adult Ugt1 KO Animals Have High Plasma and Brain Bilirubin Level 

To study alternative bilirubin clearance pathways in vivo, we used a mouse strain with genetically induced hyperbilirubinemia due to a null mutation in the Ugt1a1 gene [29,30]. Western blot analysis confirmed the absence of Ugt1a1 protein expression in the liver of adult (10–12 weeks old) homozygous Ugt1 KO mice, leading to high plasma bilirubin levels (Figure 1A,B). Determination of tissue bilirubin content using the UnaG protein, in WT and KO animals, showed very high bilirubin levels in the liver, forebrain and cerebellum of mutant mice (Figure 1C–E). 

### 2.2. CAR and Cyp2a5 Are Up-Regulated in Untreated Ugt1 Mutant Animals Compared to Wild Type Animals

Since CAR expression was upregulated in hyperbilirubinemic phenylhydrazine (PHZ)-treated WT mice [26], we first determined whether untreated Ugt1a1 KO animals also exhibited increased levels of active CAR. Western blot with the nuclear fraction of liver tissue revealed a 50% increase of CAR in mutant animals (Figure 2A), in comparison to their WT littermates. We then analyzed the expression of Cyp2a5, a member of the cytochrome P450 group of enzymes involved in the metabolism of different xeno- and endobiotics. Cyp2a5 was shown to function as hepatic “bilirubin oxidase” [14,32] and it has also been linked with higher bilirubin level [33,34]. We determined Cyp2a5 mRNA and protein levels by qRT-PCR and Western blot, respectively, and further measured its enzymatic activity by the production of the reaction end-product 7-hydroxycoumarin. We observed the up-regulation of Cyp2a5 protein expression along with up-regulated enzyme activity in the liver of KO animals, in comparison to WT animals while mRNA levels showed a trend toward up-regulation (Figure 2B,D). In addition, we also checked the gene expression levels of the peroxisome proliferator-activated receptor (PPAR) α, which belongs to the PPAR family of lipid sensors. These ligand-activated transcription factors regulate lipid and fatty acid homeostasis, as well as energy storage and expenditure [35], and it has been linked with higher bilirubin levels [36,37]. We observed the up-regulation of PPAR α both in untreated Ugt1^−/−^ liver mRNA and in mutant mice treated with TCPOBOP (Appendix A). 

### 2.3. CAR Activation Decreases Plasma Bilirubin Level in Ugt1^−/−^ Mice

Next, to determine its effect in the secondary bilirubin degradation pathways in the absence of Ugt1a1 activity, we administered mutant mice with phenobarbital (PhB), a compound used to increase Ugt1a1 transcription in CNS2 patients [38,39], which is also a known activator of CAR [40]. We observed around a 40% decrease in plasma bilirubin level following PhB treatment (Appendix A). This was associated with the transcriptional upregulation of CAR, PXR and AhR nuclear receptors (Appendix A), as well as CAR-target genes and bilirubin transporters including Cyp2B10, Cyp2c55, Gstm3, Cyp2a5, and ABC transporters Abcb1, Abcc1, Abcc2, and Abcc3 (Appendix A). 

We hypothesized that the marked reduction in plasma bilirubin could have been mediated by the activation of CAR-target genes. Thus, we treated mutant animals with TCPOBOP, a stronger CAR activator, and assessed bilirubin levels and the expression levels of CAR-target genes. Male Ugt1 mutant mice of 10–12 weeks of age were treated intraperitoneally with TCPOBOP once per day for three consecutive days, and animals were sacrificed on the fourth day, 24 h after the last injection (Figure 3A). TCPOBOP treatment resulted in about a 38% decrease in blood plasma bilirubin levels, compared to the untreated mutant animals (Figure 3B and Appendix A). Liver tissue bilirubin level was also decreased by 40% after TCPOBOP treatment. However, despite the important decrease in plasma bilirubin levels, TCPOBOP treatment did not show any significant effect on forebrain and cerebellar bilirubin levels (Figure 3D,E). The body weight of TCPOBOP-treated animals showed a non-statistically significant reduction during the treatment (Appendix A), while we observed an increase in the liver/body weight ratio and liver weight (Appendix A). However, the assessment of alanine aminotransferase (ALT) levels showed no statistically significant difference (Appendix A), indicating that TCPOBOP administration did not cause liver damage. 

### 2.4. TCPOBOP Treatment Activated CAR and Induced CAR-Target Genes

We next analyzed gene expression levels of CAR and CAR-target genes in liver tissue from Ugt1 mutant mice treated with TCPOBOP, using qRT-PCR. We observed that TCPOBOP treatment significantly induced the major targets: CAR, Cyp2b10, and Gstm3 transcription (Figure 4A). Western blot analysis further revealed significant nuclear translocation of CAR protein, with an increase in the nuclear fraction and a decrease in the cytosolic fraction, confirming the activation of the CAR protein function (Figure 4B,C)**.** We further found that pharmacological activation of CAR with TCPOBOP leads to a significant mRNA induction of drug metabolic genes such as AhR (33x), Cyp1a1 (15x), Cyp2a5 (17x), and Cyp2c55 (750x) (Figure 4D). The increase in mRNA level of Cyp2a5 by TCPOBOP was also associated with the significant up regulation of Cyp2a5 protein and enzymatic activity (Figure 4E,F). These results confirm the capacity of TCPOBOP to successfully activate alternative bilirubin clearance pathways in the absence of functional Ugt1a1, and may explain the ability of the drug to significantly reduce plasma bilirubin levels in our animal model.

### 2.5. TCPOBOP Treatment Up-Regulated Transcription of Bilirubin Transporter Genes

Following up on our findings that treatment with xenobiotic chemicals activates key transcription factors in the liver, we analyzed the induction of hepatic drug disposition genes in our experimental setup. Coordinated regulation of Phase I, Phase II, and transporter genes were observed in the liver of Ugt1 null mice after TCPOBOP treatment. We analyzed liver genes, which are important in the transport of bilirubin and bilirubin glucuronides between blood and hepatocytes. Specifically, we observed the transcriptional up-regulation of Abcc1(Mrp1), Abcc2 (Mrp2), Abcc3(Mrp3), Abcc4 (Mrp4), Abcc5 (Mrp5), Abcc6 (Mrp6), Abcb1 (Mdr1), and Slco1b2 (Oatp1b2), following TCPOBOP treatment (Figure 5A). 

### 2.6. TCPOBOP Treatment Activated Nrf2-Related Genes 

It has been reported that oxidative stress-induced Nrf2 activation is associated with a decrease in blood bilirubin levels [28]. We herein evaluated Nrf2 activation and the oxidative status of the liver upon TCPOBOP treatment in our mutant model. We measured ROS levels in the liver and performed the qRT-PCR analysis of Nrf2 and ARE sensitive genes (Nqo1, HO1, Gsta1, and Gsta2) of liver mRNA extracted from the treatment groups. We observed increased ROS generation in the TCPOBOP-treated Ugt1 KO mice compared to untreated control (Figure 5B), and found that TCPOBOP treatment significantly up-regulated Nrf2, Nqo1, HO1, Gsta1 and Gsta2 mRNA expression (Figure 5C).

### 2.7. Liver Microsomes from TCPOBOP Treated Ugt1a1 KO Mice Show Increased Ability to Degrade Bilirubin In Vitro

Liver microsomes act as the house for Phase I and Phase II metabolism enzymes. We next investigated whether TCPOBOP treatment results in an increase in bilirubin degradation in vitro. We performed an in vitro bilirubin degradation assay with isolated microsomes. We first performed an in vitro experiment to activate CAR in the mouse hepatic cell line NMuli with TCPOBOP. This treatment resulted in up-regulation in the expression of CAR and Cyp2a5 genes (Appendix A). The bilirubin disappearance assay with the isolated microsomes from TCPOBOP-treated cells resulted in an increased ability to degrade bilirubin (~2 folds) in comparison to the untreated cells (Appendix A). For the blank, the difference in absorbance for the sample without NADPH was used to make sure that the degradation was Cyp enzyme-dependent only. 

For in vivo studies, we isolated liver microsomes from untreated and TCPOBOP-treated Ugt1 KO mice. Our results suggested that the microsomes from TCPOBOP-treated livers harbor around 40% increased capacity to degrade bilirubin in the absence of Ugt1a1 activity, in comparison to the untreated and vehicle treated controls (Figure 5D). 

## 3. Discussion

Hyperbilirubinemia is normally managed by intensive phototherapy, but in some cases patients are unresponsive to the treatment and risk permanent brain damage and death. In addition to the well-known hepatic glucuronidation and excretion, and the direct, non-enzymatic oxidation of bilirubin [41], alternative pathways have been reported for the clearance of bilirubin [16,17,41,42]. However, these alternative mechanisms have been mostly studied in cell systems or in animals, in the presence of endogenous Ugt1a1 activity and their effective role in bilirubin disposal remains to be fully elucidated. Thus, to shed light on the role of these alternative pathways in the disposal of bilirubin under in vivo conditions, we took the advantage of an animal model bearing a null mutation in the Ugt1a1 gene, and consequently devoid of Ugt1a1 activity [29,30]. We performed the experiments in adult animals, an experimental condition in which all genes associated with metabolism and clearance pathways should be well expressed [43]. 

We have found up-regulated activity of Cyp2a5 and an activated expression of CAR in the liver of untreated adult Ugt1 KO animals in comparison to WT mice. This could be associated with the prolonged exposition of these animals to high levels of bilirubin [26,33]. Increased production of bilirubin in obese mice resulted in the elevation of the fat burning nuclear receptor, PPARα [36]. In addition, bilirubin has been shown to directly bind to activate PPARα and its downstream genes [37]. We have found that PPARα expression is higher in Ugt1^−/−^ animals (~1.6 folds) in comparison to the WT animals. PPARα activation has been shown to enhance CAR expression in rat hepatocytes [35]. Similarly, TCPOBOP treatment, such as other xenobiotics, has been reported to elevate PPARα expression [44,45]. Treatment of Ugt1^−/−^ animals with TCPOBOP further up-regulates PPARα expression (~2.5 folds) (Appendix A).

One of the main players in regulating alternative bilirubin clearance pathways is CAR, the xenobiotic sensor nuclear receptor. CAR is a key transcription factor for different target genes implicated in hepatic detoxification pathways, thereby protecting the liver from bile acid–mediated injury [21,26,46,47,48]. Thus, being the central player in the regulation of different molecular pathways associated with bilirubin metabolism [26,49], we hypothesized that CAR could be a plausible target to activate bilirubin detoxification. Under the non-stimulated conditions, CAR is present in the cytoplasm but, in response to specific stimuli, it is dephosphorylated and translocates into the nucleus [50], binding the phenobarbital-responsive enhancer modules (PBREM) in the promoter region of certain genes, such as Cyps and Ugt1a1, activating gene transcription [51,52] and, consequently, enhancing bilirubin clearance [21,26]. Cyp2a5, the mouse orthologue of human Cyp2a6, has been reported to be induced by bilirubin and other xenobiotics [32,34]. Our results suggest that the increase in bilirubin clearance in the plasma and liver could be related to the up-regulation of Cyp genes (Figure 4), as bilirubin is also degraded by Cyps, particularly Cyp1 and Cyp2a5 [16,17,32]. Several transcriptional factors along with CAR have been found as candidates for Cyp2a5 up-regulation [34,49].

Phenobarbital (PhB) is clinically used for CNS patients with residual Ugt1a1 activity, as it increases hepatic bilirubin metabolism by increasing Ugt1a1 expression [53]. PhB is a known CAR activator [40], and the treatment of Ugt1^−/−^ animals with PhB resulted in a decrease in serum bilirubin level in the present study (Appendix A). These results encouraged us to further investigate the bilirubin detoxification mechanism with a more potent CAR activator in the absence of Ugt1a1 activity. We found that treatment of mutant mice with TCPOBOP, the most potent CAR activator [54], showed even better effects on gene activation than PhB in the WT animals (Appendix A). Our findings indicate that TCPOBOP treatment significantly up-regulated a number of genes that are key for bilirubin transport and metabolism in the mouse liver. Upon treatment with TCPOBOP, we recorded about a 38% decrease in blood plasma bilirubin level in Ugt1^−/−^ animals when compared with the untreated mutant mice of the same age. Interestingly, the decrease in plasma bilirubin level was observed only on day 4 of the TCPOBOP treatment (Appendix A). We have also observed a similar reduction in liver tissue bilirubin in the treated KO group. 

Bilirubin transporters belong to the family of ABC transporters (ATP-binding-cassette transporters). They participate in the binding and transport of different biological substances and drugs [55]. Mostly, the transportation of CB into the bile is mediated by Abcc2. Along with the transport of a fraction of CB, Abcc3 has been also involved with the drug metabolism of PhB and TCPOBOP [56]. Upon treating mutant mice with TCPOBOP, we observed increased gene expression of ABC transporters that are involved in bilirubin transport. Our previous work showed the involvement of Abcb1 in the efflux of unconjugated bilirubin from the cerebellum [57]. Here, we observed more than 20- and 10-fold up-regulation of liver Abcb1 and Abcc1, respectively, together with the up-regulation of the other transporters tested, following TCPOBOP treatment. 

The transcription factor Nrf2, a sensor of oxidative stress [58], is a key player in the antioxidant defense system, as it can induce the transcription of antioxidant and cytoprotective genes through its interaction with the Antioxidant Response Elements (ARE). Nrf2 dependent genes include the Phase II group of drug metabolizing enzymes such as NAD(P)H:quinone oxidoreductase 1 (Nqo1), glutathione S-transferase (Gsta), Glutathione Reductase (GR), and Heme-Oxygenase (HO1). These members participate in the detoxification and cytoprotective processes through the elimination of different endo and xenobiotic elements, by reduction and conjugation reactions [59,60]. The PBREM cluster of the Ugt1a1 gene houses the binding sites for Nrf2 along with CAR [15,61,62]. It is quite predictable that the activation of Nrf2 could also play a functional role in the activation of Ugt1a1. Yoda et al. have reported that Phenethyl isothiocyanate (PEITC) treatment in hUgt1a1 mice acts dually by activating both Nrf2 and CAR for the induction of Ugt1a1 [28]. However, this mechanism will not result in increased bilirubin clearance in the complete absence of Ugt1a1 activity, as in our model and in the CNSI patients. Thus, alternative mechanisms need to be activated in the absence of Ugt1a1 activity to facilitate bilirubin clearance in this system. Alternate pathways of oxidative degradation of bilirubin are present under certain conditions [16,17,32] and they are majorly performed by the Cyps. However, their activation is probably not efficient enough for the survival of hyperbilirubinemic neonates in the absence of Ugt1a1 activity. This limitation may be explained by the natural delayed expression of phase I and phase II enzymes after birth [63] and by the observed delayed action on bilirubin clearance upon the treatment (Appendix A).

From the qRT-PCR analysis of the genes from liver tissue, we found that TCPOBOP treatment highly up-regulated liver cytochrome P450 monooxygenases (Cyps). Among the genes involved in UCB catabolism, Cyps are the most likely molecules to play an important role in the elimination of bilirubin in the absence of Ugt1a1 [16,17,18,32,42]. Cyps are the Phase I group of enzymes that are responsible for the metabolism and oxidation of several endogenous and xenobiotic substances [64] residing within the microsomal compartment of the cell. We hypothesized that upon TCPOBOP-mediated induction of Cyps, the microsomal fraction should present an increase in its capacity to oxidize bilirubin in vitro. We observed around a 40% increase in bilirubin degradation rate in liver microsomes isolated from TCPOBOP-treated Ugt1a1 KO mice in comparison to the untreated animals. This result is in line with the observed decrease in the plasma and liver tissue bilirubin decrease level following TCPOBOP treatment, suggesting that decreases in plasma and liver bilirubin may be mainly related to the increased activity of Cyps and not directly related to the observed up-regulation of ABC transporters. 

In spite of the drop in plasma and liver bilirubin levels, and the suggested potential of brain cytochrome P450 enzymes in lowering tissue bilirubin [65], TCPOBOP treatment had no significant impact on lowering brain (forebrain and cerebellum) bilirubin levels. Since the TCPOBOP treatment did not reduce plasma bilirubin levels until day 4 of the treatment, a longer treatment duration may be necessary to obtain a significant reduction in brain bilirubin levels. Importantly, although plasma ALT level was unaltered, TCPOBOP treatment resulted in hepatomegaly in these animals. These findings suggest that other less toxic compounds may be explored for a longer and/or chronic regimen. In addition, prolonged use of CAR activators and certain inducers of Nrf2 could exert carcinogenic effects [66,67]. For these reasons, special attention and thorough investigation are required for the potential use of these types of treatments. 

In the present study, we provided the proof-of-principle that the activation of alternate bilirubin clearance pathways could, at least, partially improve the hyperbilirubinemic condition resulting from the complete absence of Ugt1a1 activity in vivo. TCPOBOP, being a potent CAR inducer, significantly activates several hepatic genes that are involved in bilirubin transport and metabolism. The overall decrease in blood plasma and liver tissue bilirubin levels (both around 40%) could result from the synergistic effect of hepatic activation of an array of different genes. However, brain bilirubin levels showed no effect of the treatment, suggesting that a longer treatment duration or higher reduction in plasma bilirubin may be required. Altogether, our findings indicate that activation of alternate pathways may not be efficient enough for lowering the bilirubin level close to WT level and may not be effective in acute situations due to the delayed response. CAR activation could be beneficial in a combinatorial therapeutic strategy with other associated treatments, such as PT. Further mechanistic studies with adult and neonatal animals could be helpful to understand the situation and the molecular interplay to a better extent.

## 4. Materials and Methods

### 4.1. Animals and Drug Treatment

Mice were housed and handled according to the institutional guidelines. Ethical and experimental procedures were reviewed and approved by the ICGEB board, with full respect to the EU Directive 2010/63/EU for animal experimentation, and by the Italian Ministry of Health (Protocol code DGSAF0024706, date of approval 11 February 2014). All experiments involving animals were conducted in full respect of the ARRIVE 3R principles. Ugt1 KO mice with an FVB/NJ background were generated as previously described [29,30]. Mice were kept in a temperature-controlled environment with a 12/12 h light/dark cycle. They received a standard chow diet and water ad libitum. Wild type animals used in this study were from FVB/NJ genetic background, and were obtained from the Het x Het crosses for the Ugt1a1 locus. 

To assess the effect of CAR activation, the animals were treated with the specific CAR agonist TCPOBOP (1,4-bis-[2-(3,5,-dichloropyridyloxy)]benzene) [31]. In this study 10-to-12 week old male Ugt1 KO mice (*n* = 3 to 4) were injected intraperitoneally (i.p.) either with TCPOBOP (Sigma-Aldrich, St. Louis, MO, USA) at 3 mg/kg/day or with corn oil (vehicle control), for 3 consecutive days. On day 4, animals were sacrificed under isoflurane anesthesia. Blood samples were collected by intracardiac puncture and the liver was promptly excised. A portion of each liver was kept in 10% neutral formalin for histological examination and the remainder was washed in PBS (pH 7.4), snap-frozen in liquid nitrogen and stored at −80 °C for other studies. For Tissue bilirubin measurement, animals were transcardially perfused with PBS, until the liver was cleared of blood. Then, the target organs were stored at −80 °C for further studies.

### 4.2. Cell Culture and Treatment

Murine hepatic cell line NMuli (ATCC CRL-1638) was cultured in DMEM media supplemented with 10% FBS. Cells were treated with 1 µM TCPOBOP [31] for 24h for the activation of CAR and Cyp2a5. Dose- and time-dependent studies were performed to find out the effective dose for TCPOBOP treatment. Post treatment, the cells were collected and processed for RNA isolation and cDNA synthesis. For Bilirubin disappearance assay, microsomal fraction was isolated from the cellular homogenate.

### 4.3. Biochemical Analysis of Plasma Samples 

Determination of total plasma bilirubin in the different treatment groups of animals was performed as previously described [29]. The level of plasma liver injury marker alanine aminotransferase (ALT) was determined by Abnova ALT Assay Kit (Abnova, Taipei, Taiwan).

### 4.4. mRNA Extraction and qRT-PCR Analysis

Total RNA from liver tissue was extracted using TRI Reagent (Thermo Fisher Scientific, Waltham, MA, USA) and cDNA was prepared from 1 μg RNA using M-MLV Reverse Transcriptase (Invitrogen), according to the manufacturer’s instruction. qRT-PCR was performed on C1000 Thermal Cycler CFX96 (Bio-Rad, Hercules, CA, USA) using the iQ SYBR Green Supermix (Biorad). Expressions of the gene of interest were normalized to the mouse albumin as the housekeeping gene, unless indicated otherwise. All the primers used in this study are listed in Table 1.

### 4.5. Protein Extraction and Western Blot Analysis

Liver tissue was homogenized in RIPA lysis buffer (Cell Signaling Technology Danvers, MA, USA) to prepare whole cell lysate. Cytosolic and nuclear fractions were prepared according to the standard protocol [68]. All the buffers were supplemented with proteases (Complete Mini protease inhibitor, Roche, Basel, Switzerland) and phosphatase inhibitors (PhosSTOP, Roche. Basel, Switzerland).

An equal amount of protein (30 μg) for each sample was resolved on SDS–PAGE and electrophoretically transferred to nitrocellulose or PVDF membranes. The membrane was blocked with 5% non-fat milk in PBST for 2 h and then incubated with specific primary antibodies for CAR (NR1I3, Origene, Rockville, MD, USA), Ugt1 (Boster, Pleasanton, CA, USA) and Cyp2a5 (Abcam, Cambridge, UK) overnight at 4 °C and subsequently exposed to HRP-conjugated secondary antibodies. The bands were visualized with ECL detection. HSP70 (Enzo, New York, NY, USA) was used as the whole cell loading control. Tubulin (Santa Cruz, Dallas, TX, USA) and Histone H3 (Cell Signaling Technology, Danvers, MA, USA) were used as the loading control for the cytosolic and the nuclear fraction, respectively.

### 4.6. Tissue Bilirubin Measurement

Tissue Bilirubin level was measured with a bilirubin-specific fatty-acid binding protein, UnaG [69]. The method using this bilirubin-dependent UnaG fluorescence to measure the tissue bilirubin content was adapted from Adeosun et al. [70]. Briefly, liver tissue samples were homogenized with RIPA lysis buffer (supplemented with protease and phosphatase inhibitor). Tissue homogenate was centrifuged at 13,200 rpm and the supernatant was collected to determine the protein concentration with standard procedures.

UnaG protein was expressed in BL21 (DE3) cells, containing the UnaG plasmid (pET-32a(+)-UnaG), containing the Ampicillin selection marker. The overnight 6 mL starter culture was transferred into 500 mL ampicillin-supplemented culture medium and incubated with shaking. The growth was monitored every 30 min until OD_600_ of 0.4–0.6 was reached. The culture was then induced with IPTG at a final concentration of 1.5 mM. Bacterial cells were lysed with RIPA buffer (supplemented with protease and phosphatase inhibitor) (10 mL for the pellet from 500 mL culture). The crude protein lysate of the bacteria was used to detect bilirubin concentration in samples and standards. We used 200 μg of tissue lysate and 20 μL of UnaG lysate for each reaction tube. The total reaction volume was adjusted to 500 μL with the assay buffer (100 mM Tris Base, 1 mM EDTA in ddH_2_O, pH 8.7). Two hundred microliters of each of the reactions, blank and standards, were loaded in duplicate in 96-well black-transparent flat-bottom plate. Bilirubin standards of 0, 10, 100, 1000, and 10,000 nM were used to plot the standard curve. The blank reaction was also set with UnaG, RIPA, and assay buffer. Bilirubin standards were run together with the samples and blank in all determinations.

The fluorescence reading was taken using Envision multilabel reader (Perkin-Elmer, Buckinghamshire, UK) with λ ex = 485 nm and λ em = 528 nm. The readings were repeated after 30 min and were tracked up to 3 hr. The quantity of bilirubin in the samples was expressed as nM of Bilirubin per μg of Protein.

### 4.7. Microsome Preparation

Microsomes were prepared as described previously [30] with some modifications. The tissue was homogenized at 4 °C in homogenization buffer (1.15% KCl in 10 mM Potassium phosphate buffer pH 7.4). Homogenate was centrifuged at 2000× *g* for 10 min at 4 °C and the supernatant was collected. The supernatant was centrifuged at 9000× *g* for 10 min at 4 °C to remove the nuclei and further centrifuged at 100,000× *g* for 1 h at 4 °C to precipitate the microsomal pellet. The microsomal fraction was re-suspended in microsomal buffer (50 mM Tris-HCl pH 7.4, 10 mM MgCl_2_ at 4 °C) and protein concentration was determined by the Bradford method. All the buffers were supplemented with protease and phosphatase inhibitor (cOmplete Protease inhibitor cocktail, Roche, Basel, Switzerland; phosSTOP, Roche, Basel, Switzerland).

### 4.8. Bilirubin Disappearance Assay

The bilirubin disappearance assay was performed as described by Kim et al. [33]. Briefly, microsomes (1 mg/mL final protein concentration) were mixed with incubation buffer (0.1 M Tris–HCl pH 8.2, 26 mM KCl, 2 mM EDTA) in a total volume of 200 μL and placed in 96-well plates in triplicate. After 5 min of pre-incubation, NADPH (2.0 mM) was added to both sample and the control wells, and blank absorbance measurements were taken at 450 nm (Envision multilabel reader, Perkin-Elmer, UK). Bilirubin in DMSO (10 μM final concentration) was immediately added to the sample plate and absorbance at 450 nm was recorded for three cycles at 10 min/cycle. The rate of bilirubin disappearance was expressed as pmol Bilirubin degradation/hr/mg protein, using a ε = 72.692 cm^−1^ mM^−1^, which was obtained experimentally under the conditions of the assay. Additional wells containing the samples without NADPH were also run to control for the spontaneous disappearance of bilirubin.

### 4.9. Coumarin 7-Hydroxylase Activity Assay

CYP2A5 activity was determined by assessing 7-hydroxylase activity towards coumarin as described by Aitio et al. [71] with some modifications. Briefly, 100 μg (~200 μL) of liver microsome was added to 300 μL of substrate buffer (50 μL of 1 M potassium phosphate buffer pH 7.4 with 5% Glycerol, 25 μL of 0.1 M MgCI_2_, 5 μL of 10 mM Coumarin (Sigma-Aldrich), and 220 μL ddH_2_O). Before starting the reaction, 100 μL of 7.5 mM reduced nicotinamide adenine dinucleotide phosphate (NADPH, Sigma-Aldrich) was added. The reaction mixture was incubated for 20 min at 37 °C in a shaking water bath. The reaction was terminated by the addition of 500 μL 2N HCl (Merck, Kenilworth, NJ, USA). The solution was centrifuged at 500× *g* for 20 min at 4 °C to remove cellular debris.

The amount of hydroxylated coumarin was determined by fluorescence using Envision multilabel reader (Perkin-Elmer, Buckinghamshire, UK) with λ ex = 390 nm and λ em = 440 nm. Serial dilutions of Umbelliferone (7-hydroxycoumarin) at 1, 0.5, 0.25, 0.125, 0.0625 μM were used for the standard curve. The sample fluorescence emissions were interpolated from the standard curve. The amount of 7-hydroxycourmarin was expressed as μmol 7-hydroxycoumarin/mg microsome protein/minute.

### 4.10. Reactive Oxygen Species (ROS) Quantification in Liver Tissue

Tissue ROS production was estimated by using 2,7-dichlorofluorescein diacetate (DCF-DA) as a probe, according to the method of Gabbia et al. [72], with some modifications. Briefly, 200 mg of liver tissue was homogenized in 2 mL of ice-cold Tris-HCl buffer (40 mM, pH = 7.4). One hundred microliters of tissue homogenate was incubated with the assay media (20 mM Tris–HCl, 130 mM KCl, 5 mM MgCl_2_, 20 mm NaH_2_PO_4_, 30 mM glucose, and 10 μM DCF-DA) at 37 °C for 40 min. As a control for tissue auto-fluorescence, 100 μL of tissue homogenate was incubated with 1 mL of incubation buffer, without DCF-DA under similar conditions. The fluorescence intensity (λ ex = 485 nm and λ em = 525 nm) of the samples was assessed using Envision multilabel reader (Perkin-Elmer, UK). ROS readings were expressed as arbitrary Fluorescence Intensity Units.

### 4.11. Statistical Analyses

The Prism package (GraphPad Software, La Jolla, CA, USA) was used to analyze the data. The results are expressed as mean ± S.D. The values of *p* < 0.05 were considered statistically significant. Depending on the experimental design, Student’s *t*-test or one way ANOVA, with Bonferroni’s post hoc comparison tests, were used, as indicated in the figure legends and text. 

## Figures and Tables

**Figure 1 ijms-23-10703-f001:**
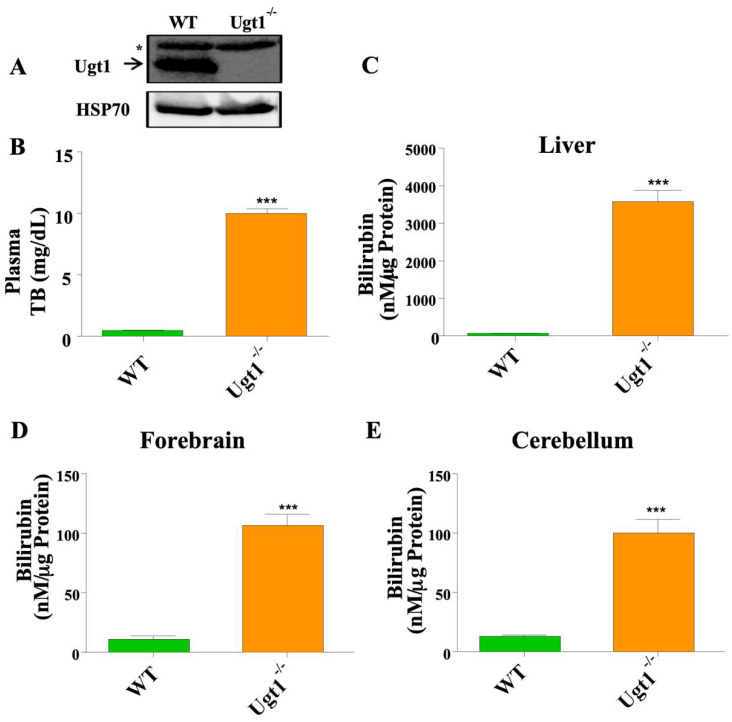
**Adult Ugt1**^−/−^**animals have an upregulated level of blood and tissue bilirubin.** (**A**) Western blot analysis for Ugt1a1 expression in the liver of WT and Ugt1^−/−^ adult mice. Hsp70 was used as the loading control. (**B**) Plasma total bilirubin level (mg/dL) in WT (*n* = 5) and Ugt1^−/−^ (*n* = 5) adult mice. Tissue bilirubin level (nM/μg Protein) in (**C**) Liver (**D**) Forebrain and (**E**) Cerebellum in WT (*n* = 3) and Ugt1^−/−^ (*n* = 3 for each treatment group) adult mice. Values represent mean ± SD. *t*-test was performed and *** *p* < 0.001 were interpreted as statistically significant. The asterisk in Panel (**A**) indicates an unspecific band.

**Figure 2 ijms-23-10703-f002:**
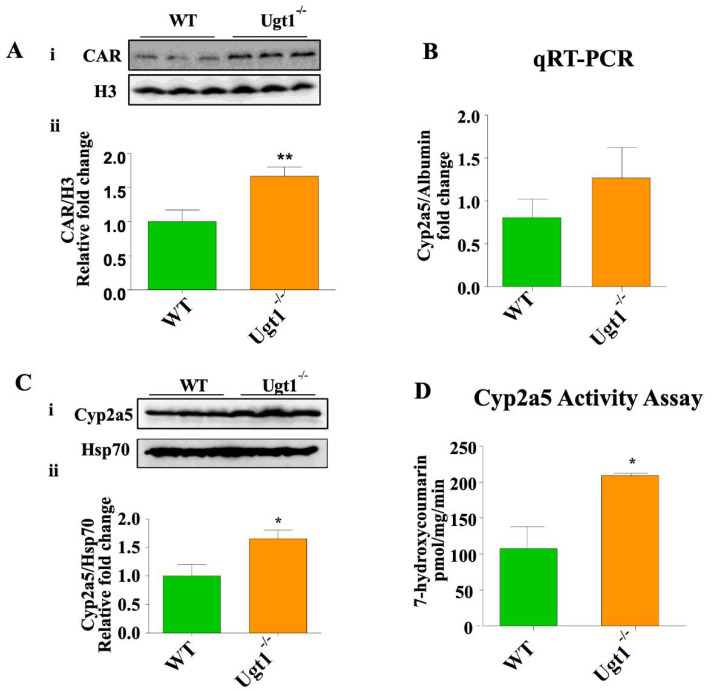
**CAR and Cyp2a5 expression in the liver of WT and Ugt1**^−/−^**animals.** (**A**) **i.** Western blot analysis of nuclear CAR expression in the liver of WT and Ugt1^−/−^ adult mice. Histone H3 used as a nuclear loading control. **ii.** Quantification of the Western blot analysis. (**B**) qRT-PCR analysis of Cyp2a5. Albumin was used as the internal control. (**C**) **i.** Western blot analysis for Cyp2a5 expression from total liver protein extract of WT and Ugt1^−/−^ adult mice. Hsp70 was used as the loading control. **ii.** Quantification of Western blot analysis. (**D**) Liver Cyp2a5 activity assay in WT and Ugt1^−/−^ adult animals. Values represent mean ± SD. *t*-test was performed. * *p* < 0.05 and ** *p* < 0.01 were interpreted as statistically significant.

**Figure 3 ijms-23-10703-f003:**
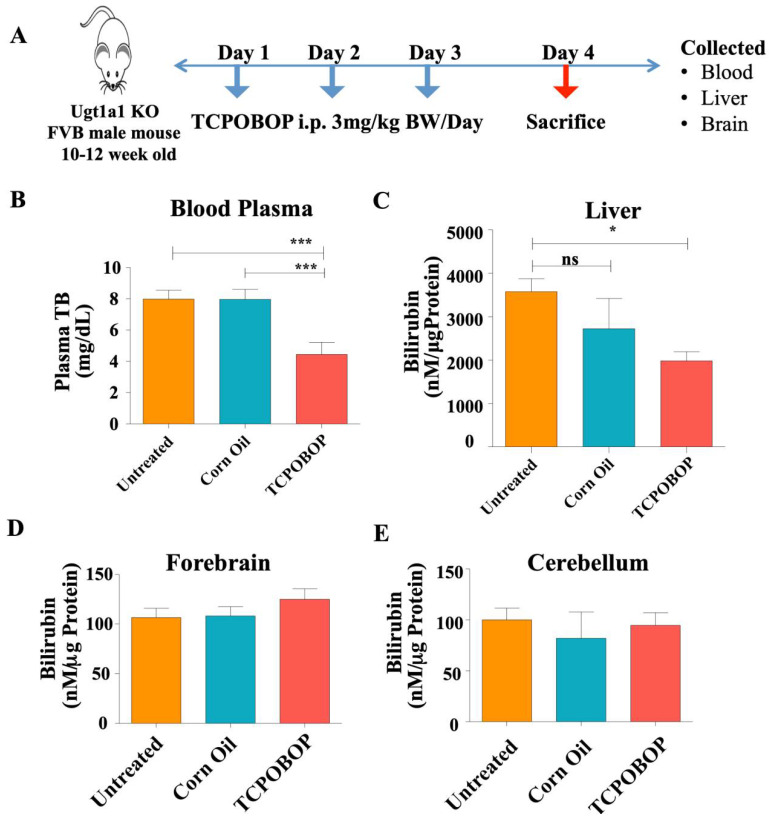
**TCPOBOP treatment in adult Ugt1**^−/−^**animals.** (**A**) Schematic diagram of the experimental design of TCPOBOP treatment in Ugt1^−/−^ adult animals. (**B**) Plasma total bilirubin level (mg/dL) in different treatment groups (*n* = 6 each group). Tissue bilirubin level (nM/μg protein) in (**C**) Liver (**D**) Forebrain, and (**E**) Cerebellum in Untreated, Corn oil (vehicle control), and TCPOBOP-treated groups (*n* = 3 for each treatment group). Values represent mean ± SD. One way ANOVA test was followed by Bonferroni’s multiple comparison test. *** *p* < 0.001, and * *p* < 0.05 were interpreted as statistically significant. The treatment groups were: Untreated, Corn Oil (vehicle control), and TCPOBOP. ns = non-significant.

**Figure 4 ijms-23-10703-f004:**
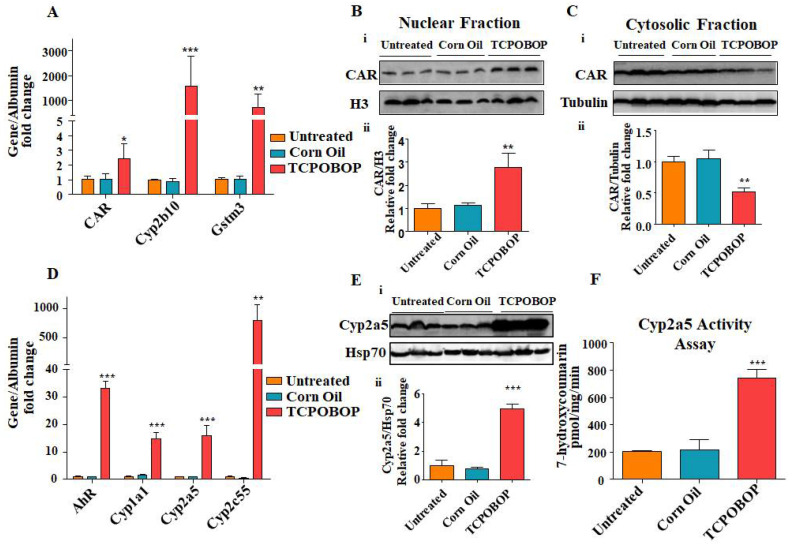
**TCPOBOP treatment induces gene expression of CAR and CAR-regulated genes in Ugt1**^−/−^**animals**. (**A**) qRT-PCR analysis of CAR, Cyp2b10 and Gstm3 in different treatment groups. Albumin was used as the internal control. (**B**) **i.** Western blot analysis of CAR in liver nuclear fraction extracts from different treatment groups. Histone H3 was used as the nuclear loading control. **ii.** Quantification of Western blot analysis. (**C**) **i.** Western blot analysis of CAR in liver cytosolic fraction extracts from different treatment groups. Tubulin was used as the cytosolic loading control. **ii.** Quantification of Western blot analysis. (**D**) qRT-PCR analysis of AhR, Cyp1a1, Cyp2a5, Cyp2c55 in different treatment groups. Albumin was used as the internal control. (**E**) **i.** Western Blot analysis of Cyp2a5 in liver protein extracts from different treatment groups. Hsp70 was used as the internal loading control. **ii.** Quantification of Western blot analysis. (**F**) Cyp2a5 activity assay in the liver in different treatment groups. All panels, values represent mean ± SD. One way ANOVA test was followed by Bonferroni’s multiple comparison test. *** *p* < 0.001, ** *p* < 0.01, and * *p* < 0.05 were interpreted as statistically significant (*n* = 3 for each treatment group). The treatment groups were: Untreated, Corn Oil (vehicle control), and TCPOBOP.

**Figure 5 ijms-23-10703-f005:**
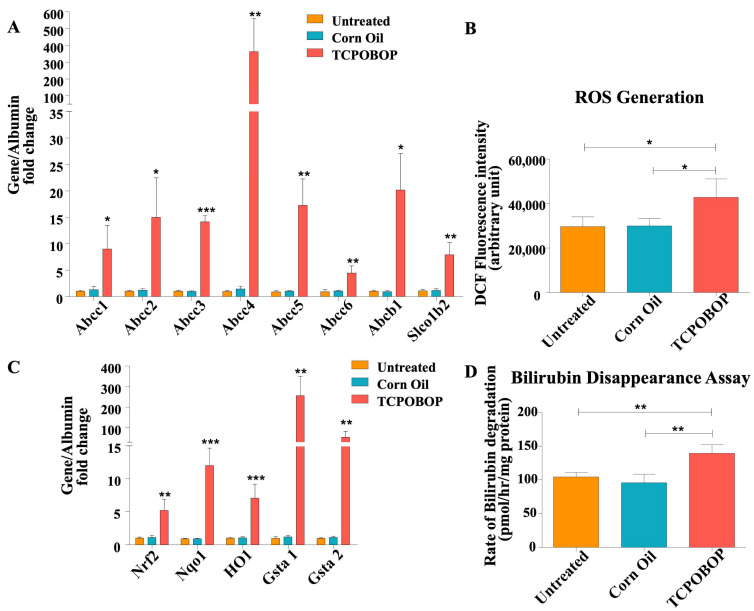
**TCPOBOP treatment activates genes involved in bilirubin metabolism and transport, and activates oxidative stress-induced signaling pathways in adult Ugt1**^−/−^**animals.** (**A**) qRT-PCR analysis of Abcc1, Abcc2, Abcc3, Abcc4, Abcc5, Abcc6, Abcb1, Slco1b2 in different treatment groups. Albumin was used as the internal control (*n* = 3 for each treatment group). (**B**) ROS generation in liver tissue in different treatment groups was determined using DCF-DA. *n* = 4. (**C**) qRT-PCR analysis of Nrf2, Nqo1, HO1, Gsta1, Gsta2 in different treatment groups. Albumin was used as the internal control. (*n* = 3 for each treatment group). (**D**) Bilirubin disappearance assay using liver microsomes from different treatment groups. Isolated microsomes were incubated with 10 μM bilirubin and the degradation rate was expressed as pmol/hr/mg protein (*n* = 4 for each treatment group). Values represent mean ± SD. One way ANOVA test was followed by Bonferroni’s multiple comparison test. *** *p* < 0.001, ** *p* < 0.01, * *p* < 0.05 were interpreted as statistically significant. The treatment groups were: Untreated, Corn Oil (vehicle control), and TCPOBOP.

**Table 1 ijms-23-10703-t001:** Primer sequences for qRT-PCR.

	Forward (5′-3′)	Reverse (5′-3′)
Albumin	GCATGAAGTTGCCAGAAGAC	TCTGCATACTGGAGCACTTC
Cyp2a5	TGCGCTATGGCTTTCTGTTG	GAAACTTGGTGTCCTTGGTG
Cyp1a1	TCATTCCTGTCCTCCGTTAC	ACATTGGCATTCTCGTCCAG
HO 1	ACAGGGTGACAGAAGAGGCTAAGAC	ATTTTCCTCGGGGCGTCTCT
Nqo1	TTTAGGGTCGTCTTGGCAAC	GTCTTCTCTGAATGGGCCAG
CAR	TGCAGGTTGCAGAAGTGTCTA	GGACCAGTTCTTTCTGCTGC
Cyp2b10	GTTCCACACAGCATAACCAC	CCTTGTGGGCTTCTTGTTAC
Cyp2c55	CGCATTCAAGAGGAAGCATC	ACAGATCACATTGGAGGGAG
Gstm3	TCCGTGTGGATACTTTGGAG	ATGGCCTTCAAGAACTCTGG
Nrf2	GGCAGAGACATTCCCATTTG	AAACTTGCTCCATGTCCTGC
Abcc1	TGATACAGCTTGAACGGAGG	AGCACAAGGGTGAAGTACAG
Abcc2	TGGTTCCTGTCCATGTTCTG	TCTGGAAACCATAGGAGACG
Abcc3	TTACCTGAGACACCATCAGC	CCAAGGGTGTGACAAAGAAG
Abcc4	TTCTTCTGCCTCTGCAAAGC	ACGATTTCTCCCACGCATAC
Abcc5	TGAATGAAGTTGGGCCAGAC	TGTACTCCAAGAGGTGCTTC
Abcc6	TCCTAGCTGTGCTGATGTTC	TGGACTTTCTGGAACCACTG
Slco1b2	GAAACAAGCTCCTGCCAACC	GTCTGACCAAACTGCTGCTC
Abcb1	GATAGGCTGGTTTGATGTGC	TCACAAGGGTTAGCTTCCAG
Gsta1	AGCCCGTGCTTCACTACTTC	TCTTCAAACTCCACCCCT GC
Gsta2	AATCAGCAGCCTCCCCAAT	TCCATCAATGCAGCCACACT
Ugt1a1	TCTGGCTGATGAGAAGTGACT	GAAAACAACGATGCCATGCT
AhR	GGCAGCTTATTCTGGGCTATAC	ATCAAGCGTGCATTGGACTG
PXR	GACGCTCAGATGCAAACCTT	TCTTCTCCGCGCAGCTGCA
Gapdh	ATGGTGAAGGTCGGTGTGAA	GTTGATGGCAACAATCTCCA

## Data Availability

Not applicable.

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
