# Peer review of "Activation of Alternative Bilirubin Clearance Pathways Partially Reduces Hyperbilirubinemia in a Mouse Model Lacking Functional Ugt1a1 Activity"

_ijms, 2022, doi:10.3390/ijms231810703_

Round 1

Reviewer 1 Report

Hi, 

Please see my comments attached. 

Best,

Li

Reviewer 2 Report

This is an interesting study designed to test activation of the constitutive androstane receptor (CAR) in a model of extreme hyperbilirubinemia due to absence of the UGT1A1 gene. While the authors find that activation of the CAR is able to reduce hepatic and circulating levels of bilirubin, they did not observe any significant decrease in brain levels of bilirubin in response to CAR activation in the model studied.

1) It is not clear why the author's chose to study mice at 10-12 weeks of age as opposed to 4-5 weeks if they are interested in CAR activation as a treatment for neonatal jaundice. Is it possible that more definitive results on brain bilirubin levels could be achieve if younger mice were studied?

2) Did the authors measure the levels of organic anion transport family members (OATPs) after CAR induction? These are also capable of carrying bilirubin.

Reviewer 3 Report

The manuscript by Banjeree et al. addresses an important question in the field in hopes of better understanding how to lower plasma bilirubin levels. Overall, the manuscript is well-written, and the scientific data and analysis are robust. There are comments below that could improve the overall concepts in the manuscript. 

1. Bilirubin has previously been shown to increase CAR expression but does so by not functioning as a ligand (PMCID: 153064). It has been shown to be a ligand for PPARalpha, which induces CAR expression (PMID: 18023279 and PMID: 20208394), which PMID: 20208394 also shows that PPARalpha induces Cyp2B. The manuscript will be improved by its mechanistic details if they include measurements of PPARalpha expression by either protein or mRNA in the UGT1A1 -/- and WT mice as they did for CAR in Figure 2A.  It would also strengthen the manuscript to include these in the discussion and that bilirubin activates PPARalpha, which is the only nuclear receptor shown that it directly binds. 

2. Also, albumin might not be the best internal control for immunoblotting as it is only expressed in hepatocytes in the mature liver. Many other cell types in the liver do not express albumin, and using it as the control might not be best. This is more of a minor statement and something to consider.

3. In Figure 2A, the p84 levels are higher in the Ugt1-/- mice. If a different internal control was used, the CAR relative levels might be higher. This is more of a minor statement and something to consider.

Round 2

Reviewer 1 Report

The revised manuscript was significantly improved.

Reviewer 3 Report

The authors have fulfilled the concerns of this reviewer. This is a nicely done manuscript.